# Work-Related Injuries among Insured Construction Workers Presenting to a Swiss Adult Emergency Department: A Retrospective Study (2016–2020)

**DOI:** 10.3390/ijerph191811294

**Published:** 2022-09-08

**Authors:** Ralf Dethlefsen, Luisa Orlik, Martin Müller, Aristomenis K. Exadaktylos, Stefan M. Scholz, Jolanta Klukowska-Rötzler, Mairi Ziaka

**Affiliations:** 1Department of Orthopedics, Thun General Hospital, 3600 Thun, Switzerland; 2Department of Internal Medicine, Thun General Hospital, 3600 Thun, Switzerland; 3Department of Emergency Medicine, Inselspital, Bern University Hospital, Bern University, 3010 Berne, Switzerland; 4Department of Statistics, Suva (Swiss National Accident Insurance Fund), 6002 Lucerne, Switzerland; 5Central Office for Statistics in Accident Insurance (SSUV), 6002 Lucerne, Switzerland

**Keywords:** work-related injuries, adult emergency department, Suva, construction site accidents

## Abstract

Occupational injuries are one of the main causes of Emergency Department visits and represent a substantial source of disability or even death. However, the published studies and reports on construction–occupational accidents in Switzerland are limited. We aimed to investigate the epidemiology of fatal and non-fatal injuries among construction workers older than 16 years of age over a 5-year period. Data were gathered from the emergency department (ED) of Bern University Hospital. A retrospective design was chosen to allow analysis of changes in construction accidents between 2016–2020. A total of 397 patients were enrolled. Compared to studies in other countries, we also showed that the upper extremity and falling from height is the most common injured body part and mechanism of injury. Furthermore, we were able to show that the most common age group representing was 26–35 years and the second common body part injured was the head, which is a difference from studies in other countries. Wound lacerations were the most common type of injury, followed by joint distortions. By stratifying according to the season, occupational injuries among construction workers were found to be significant higher during summer and autumn. As work-related injuries among construction workers are becoming more common, prevention strategies and safety instructions must be optimized.

## 1. Introduction

Occupational diseases have been a concern of health professionals since antiquity: Hippocrates recognized the toxicity of lead in mining workers in the 4th century BC [1]. An occupational injury is defined as any type of injury associated to the course of work, which could result from physical, biological, chemical, or psychosocial hazards [2]. Moreover, an occupational accident is described as an unexpected and unplanned occurrence arising at or in the course of work leading to death, personal injury, or disease [3].

Occupational injuries are one of the main causes of Emergency Department visits and represent a substantial source of disability or even death, especially among young adults [4,5]. Indeed, according to the International Labour Organization, 2.78 million workers suffer fatal work-related injuries and occupational diseases annually. Additionally, each year non-fatal accidents are reported in 374 million workers [6]. It is, moreover, estimated that work-related injuries lead to a global economic loss of 4% [6], making them the second-highest health-care cost in the United States [7].

In various epidemiological studies, a wide range of contributing determinants has been identified and include work-related risk factors such as type of occupation, environmental factors, limited work experience, shift work, overtime, and physical stress [8,9,10]. Moreover, non-work-related factors, as for example sociodemographics (e.g., young age group, male gender), health-related risk factors (e.g., smoking, obesity, physical inactivity, alcohol consumption), and prior medical conditions appear to be additionally associated to work-related injuries [10].

Regarding the type of occupation, the construction industry has globally the highest accident rates, making it one of the most dangerous sectors [11]. Fabiano and co-workers [12] investigated the trends of occupational injuries among temporary workers in Italian industries and found that the construction sector has one of the highest risks for work-related injuries. Similar results were reported by Unsar et al., who studied the epidemiology of occupational injuries in Turkey between 2000 and 2005. The study highlighted the construction industry as one of the most common occupations associated with accidents, being in fact the most common in terms of fatal occupational accidents [13]. This is in accordance with recent research regarding work-related fatal injuries in Italy, which has demonstrated construction as the occupation with the most fatal injuries [14,15]. Indeed, the construction industry is simultaneously dynamic and vulnerable, and highly affected by environmental and geographical factors [16,17]. Nevertheless, the interaction between professionals from different disciplines, companies, and countries generates a heterogeneous environment affecting accident rates [18,19,20]. Moreover, indicators such as working with heavy equipment, working at heights, falling objects, noise, and vibrations act cumulative to the increase of frequency and severity of accidents on construction sites [17,21]. Furthermore, previous research has highlighted that geographical disadvantages such as working in mountainous and river regions significantly affect the possibility of accidents on construction sites [17]. In addition, climate factors such as rain, wind, light levels, and high environmental temperatures have been strongly associated with a surge in both frequency and magnitude of events on construction sites [22]. Common causes of accidents among construction workers include contact with objects —especially cutting or piercing objects— falls, overexertion, exposure to hazardous materials, and electrical shock [23,24,25,26]; often several events are involved [25]. Related to the type of injury *strains and sprains*, *cuts and lacerations*, *fractures*, and *contusions* represent the most common types of injury. However, construction accidents can lead to more severe injuries as well with burns, amputations, traumatic brain injury, and lacerations of internal organs, additionally leading to fatal outcomes [23,24,25,26]. Lastly, despite significant improvements in safety performance over the past decades, health and safety training among construction workers are commonly inadequate due to the lack of appropriate procedures and guidelines [27,28].

Based on the fact that construction industry is growing rapidly, the socioeconomic influence of work-related accidents is snowballing [16]. Indeed, the construction sector plays a fundamental role in the global economy providing millions of jobs globally and represents 6% of the world’s gross domestic product, while it is estimated to increase significantly in the next years [29].

Considering that the probability of a serious accident in the construction sector is 2.5 times higher than in other industries and that globally 30–40% of these accidents are fatal, the socio-financial consequences can be devastating. In particular, accidental events at the construction site cause disabilities and affect the integrity of employees and their families, performance of the rest of the workers, and project productivity [30]. Thus, it is necessary to reduce rates of accidents on construction sites by establishing strategies and safety measures.

Despite the accuracy of collected data regarding construction accidents by Suva Insurance (Schweizerische Unfallversicherungsanstalt—Swiss National Accident Insurance Fund), very limited data evaluating patients presented to the emergency departments with a construction-accident related injury exist in Switzerland. Hence, we aimed to investigate the epidemiology of fatal and non-fatal injuries among construction-workers over a 5-year period (i.e., from January 2016 until December 2020) on a retrospective basis in order to identify current trends with regard to year, age groups, and season, and identify possible causes, risk factors, underlying mechanisms, and trauma patterns. Moreover, we focused on the occurrence of injuries as a function of demographic characteristics, such as age and gender, as a means to suggest further prevention strategies and safety instruction protocols.

## 2. Methods and Materials

### 2.1. Design

This is a retrospective, longitudinal study of all patients who were referred to the Department of Emergency Medicine for Adults of the Inselspital, Bern University Hospital, Switzerland, one of the Level 1 Trauma centers (50,000 patients per year, catchment area of 2 million inhabitants) in Switzerland, between 2016 and 2020 after a work-related accident on a construction site and who were covered by Suva Insurance (Schweizerische Unfallversicherungsanstalt—Swiss National Accident Insurance Fund).

### 2.2. Database Search Criteria

The medical report database of the Inselspital Bern (category: University Emergency Center) was searched using the following search term: “construction site” (original in German: «Baustelle»). The medical ED report of every hit in our computerized database (Ecare, Turnhout, Belgium) was then manually screened to ensure that trauma on the construction site was present.

Information found in the emergency medicine reports fitting the mentioned search term was then exported. Initially, this medical database included 1202 patients. After the application of our exclusion criteria, 397 subjects remained.

### 2.3. Recruitment of Patients

#### 2.3.1. Inclusion Criteria

All patients with a construction-related accident extracted from the cohort between 2016 to 2020 were included in the study.

#### 2.3.2. Exclusion Criteria

Patients who were not covered by Suva insurance were excluded from the study. According to the Swiss Accident Insurance Law (German: Unfallversicherungsgesetz, UVG), all employees working in Switzerland are compulsorily insured against accidents and occupational diseases. Suva is part of the Swiss social insurance system and all Swiss companies in the industry and construction sector are legally obligate to have their employees covered by Suva. Construction workers who are employed by foreign companies are not covered by Suva and therefore excluded, as well as patients who have not been employed at all, i.e., had an accident on their own, private construction site. Due to the retrospective design of our study, it is impossible to distinguish whether a patient had an accident on his own, private construction site, or was employed by a company from abroad: a questionnaire in the context of a prospective study would allow to identify and classify those patients correctly. For this reason, we have only included patients who are insured by Suva.

Moreover, we had to exclude some patients whose clinical data were not complete and thus not suitable for statistical testing. If patients had not signed the general consent by the Inselspital in order to enable scientists to use personal health-related data for research purposes they were also excluded. Children and people younger than 16 years of age were not taken into account. In total, 397 patients with sufficient clinical and demographic data were finally included (Figure 1).

### 2.4. Demographic and Clinical Data

Our data were extracted from the consilia addressed to the general practitioners that comprised detailed clinical and radiographic descriptions available in electronic form. The following demographic and clinical data were collected:**1.** **General patient data:**GenderAge**2.** **Admission and discharge data:**Route of admissionDate of admission: weekday, month, seasonTime of admission (morning/afternoon/evening/night and whether immediately after the accident to X number of days after the accident)Triage levelsTreatment area in the ER and whether trauma room treatment took placeRoute of discharge**3.** **Admission department and hospitalization duration if hospitalized**Anamnestic data:Occupation/activity performedAccident mechanismFall (height, object causing the fall, landing area) machine handling/driving a transport vehicle Working with manual instruments transport by hand (lift weight) Moving (walking, running, climbing, tripping) Contaminating substances/explosion, burn, electrical contact Being hit by a car, being run over Entrapment/impact against object Cut on an object 
Objects causing accident:TerrainMaterial extractionElectricity Machines/manual instrumentsConveyor system Means of transportation (driving vehicle, trailer)Harmful, flammable or explosive substances/gases and dustForeign body splintersHumans and animals 
Location of accident:Building construction siteRoad construction siteExcavation pit
**4.** **Clinical and preclinical data:**Injury type (simple injury, combined injury, polytrauma)
Type of injury:Wound laceration, incl. internal organs, pneumothoraxCerebral commotio, cerebral hemorrhageDistortion, contusion, crush trauma Closed or open fracture, dislocation, amputationBurn, frostbiteChemical burns, electric shock, chemical substances, etc.Infection, poisoning, irritation of mucous membranesForeign body penetration, foreign body irritation
Treatment method: Conservative, surgical, minimally invasive, death in emergency room
Injured body site: Head, neck, spine, thorax (and thoracic organs), back, abdomen (and abdominal organs), pelvis, shoulder, upper arm, elbow, forearm, wrist, hand, hip joint, thigh, knee joint, lower leg, foot


Protection material, such as helmets, safety glasses, hearing protection, or gloves were not investigated since it was not systematically mentioned in the medical reports. This query could be investigated in further (prospective) studies.

### 2.5. Statistical Analysis

The data were summarized using descriptive statistics (mean values, percentages). The statistical analysis was performed using Stata^®^ 16.1 (StataCorp, The College Station, TX, USA), which also provided the graphs used to demonstrate our results.

Parameters were classified and presented according to absolute and percentage proportions, including median values. If reasonable and possible based on the information collected, patients were subdivided and analyzed based on the accident situation, mechanism, or injured body part. Categorical variables were analyzed using the chi-square (χ^2^) test and the Fisher exact test. Group comparisons were performed using Mann–Whitney test. The threshold of significance was set at *p* < 0.05 (two-tailed).

### 2.6. Ethical Approval

Our research project used coded data and was reviewed and approved in advance by the cantonal ethics committee of Bern (b2022-00455).

### 2.7. Definitions

In order to categorize patients correctly, a few terms are explained more in detail in the following section.

**Triage levels**: Upon presentation in an emergency center in Switzerland, patients are assigned to different urgency levels according to the Swiss Triage System (Schweizerische Triage System (STS). The following definitions were extracted from the Swiss Society for Emergency and Rescue Medicine website [31].

Urgency level 1 (acute emergency, immediate treatment): Health disorder or imminent birth that may result in the death of the patient or the loss of a limb, organ, or fetus if not treated immediately.Urgency level 2 (emergency, treatment within 20 min): Health disorder that is not life-threatening but could worsen within a short time.Urgency level 3 (moderately urgent emergency, treatment within 120 min): Health disorder for which time is not a critical factor. The patient’s condition is judged to be stable at the time of arrival.Urgency level 4 (non-urgent situation): Health condition judged stable that does not actually require emergency medical therapy.**Location of accident**: Different construction sites pose different hazards for workers.Building construction site: In order to avoid potentially fatal accidents, Suva recommends different rules: e.g., securing floor openings and fall edges from a fall height of 2 m immediately, daily scaffolding checks, operating cranes in accordance with regulations, and attaching loads safely [32].Road construction site: Traffic route and civil engineering site workers are exposed to different dangers such as passing vehicles and heavy machines and loads. Sufficient visibility and safe accesses are crucial [33].Excavation pit: During trenching and excavation work, it is essential to adhere to certain safety precautions. Otherwise, life danger can quickly arise. In particular, construction workers can be buried if an embankment is created too steeply or if the ground is additionally loaded by vehicles, for example [34].

## 3. Results

### 3.1. Patient Analysis

#### 3.1.1. Age and Sex Distribution

Between 2016 and 2020, a total of 397 cases aged 16 years or older were identified in our database as having sustained work-related injuries, while working for a construction income (Table 1). In our study, patients with occupational injuries were predominantly male (98.2%). The most represented age group was 26–35 years, with 107 patients (27%, *p* < 0.001). In the comparison between the age groups, no statistical differences were observed for the following parameters: year, month, weekday, and time of consultation, triage group, route of admission, route of discharge, treatment area, type of trauma, type of injury, location of injury, mechanism of injury, and treatment method.

#### 3.1.2. Annual and Seasonal Distribution

Though our data showed a significant difference regarding the annual number of admissions (*p* < 0.001), there was no specific trend observed. More specifically, the annual number of patients varied between 67 and 89, with the fewest incidents in 2018 and the most in 2019.

In addition, our results showed that there was a significant association between the treatment area and the year of consultation. More specifically, patients treated in *Fast track*, a new treatment area for patients with less-severe injuries [35,36], in 2016 concerned 11.8% of the cases, whereas for the years 2017–2020 the percentage ranged between 30.03% and 35.4%, indicating an increase.

In the comparison between the analyzed years, no significant differences were observed for the following parameters: age groups, month, weekday, and time of consultation, triage group, route of admission, and route of discharge. For other variables, significant differences over time have been found for some categories, but these findings might be owed to multiple testing. By stratifying according to the season, occupational injuries among construction workers were found to be significantly higher during summer and autumn (*p* < 0.001).

#### 3.1.3. Time and Day of Consultation

The daily time of consultation showed two peaks between 06:00 and 12:00 (n = 148, 37.4%), and between 18:00 and 00:00, without reaching statistical significance. However, it should be mentioned that despite not being statistically significant the number of consultations between 06:00 and 12:00 was much higher in summer and autumn compared to winter and spring (194 vs. 56). The frequency of accidents from Monday to Friday was comparable; however, much less on Saturday and Sunday.

#### 3.1.4. Location and Type of Injury

The most frequently injured body parts were the upper extremities with 162 patients (40.8%), followed by the head with 136 cases (34.3%, Table 1). The third most common location of injury was the lower extremities, with 92 cases (23.2%). The hand was the most frequent location of injury in the upper extremities, with 92 patients (23.2%) (Table 1).

Wound lacerations were the most common type of injury with 190 cases (47.9%), followed by joint distortions (123 cases; 31%). Seventy-three patients had open (24 cases, 6%) and closed (49 cases, 12.3%) fractures and 32 patients (8.1%) sustained a traumatic brain injury. Penetration and irritation by foreign objects were identified as the types of injury in 71 of the cases (17.9%) (Table 2).

Figure 2 depicts the relationship between the injured body part reported and the length of hospital stay. As shown, accidents related to the neck/spine, thorax/back, abdomen, and hip resulted in more hospitalization days (approx. up to 10 days) compared to all other body parts. Within these injuries, extreme outliers were present for the neck/spine, thorax/back, and abdomen, presumably indicating the severity of injuries related to the specific body parts.

The majority of patients presented to the Emergency Department with monotrauma (332 cases, 83.6%), while 44 patients (11.1%) sustained combined injuries without life-threatening outcome and 11 patients (2.8%) polytrauma with life-threatening injuries.

#### 3.1.5. Mechanisms of Injury

As expected, the majority of the accidents occurred at the construction site (n = 337, 87.1%), whereas for 46 patients (11.9%) the exact location of the accident remained unclear.

Among the 397 occupational injuries of construction workers, the most common mechanism of injury was *falling from a height* (87 cases; 21.9%), followed by *impact against an object* (69 cases; 17.4%) and trauma due to *working with manual instruments* (68 cases; 17.1%, Table 3). To better understand the nature of the accidents among construction workers we also recorded subgroups related to the type of accidents. In 97 patients (24.4%), the injury was associated with slippery terrain, followed by accidents causing foreign body splinters (n = 70, 17.6%). Figure 3 demonstrates the relationship between the mechanism of injury and the length of hospital stay. By far, being *driven into/run over* resulted in a higher length of hospital stay and was followed by *fall*, *manual instrument work*, *entrapment bruise,* and *cut on object*. However, for *fall*, *entrapment,* and *impact against object* some extreme outliers were observed indicating that these categories of injury mechanism could result in increased hospital stay under specific circumstances. No significant differences were observed between mechanism of injury and age group.

Figure 4 summarizes the relationship between the mechanism of injury and the mode of discharge. All cases reporting *contaminated substances* as the mechanism of injury were discharged from the hospital and treated at home. The same pattern was observed for *transport by hand* and *driving a transport vehicle*, and a very similar pattern was observed for *moving*. Although for *cut on object*, *machine handling*, and *entrapment bruise,* the majority of the cases were also discharged home, for some cases (approx., above 40%) hospitalization was required. However, and in contrast to all other mechanisms of injury, in one case *entrapment bruise* led to a fatal outcome. Finally, for *driven into/run over* accidents half of the patients were discharged home, while the other half were hospitalized, suggesting that these types of accidents may result in more severe injuries.

#### 3.1.6. Treatment

The most common type of treatment was conservative (244 cases; 61.5%), whereas 85 (21.4%) and 56 (14.1%) patients received minimal invasive and operative treatment, respectively. Patients treated conservative in 2016 concerned 50% of the cases, whereas for the years 2017–2020 the percentage ranged between 62.2% and 66.3% indicating an increase.

#### 3.1.7. Admission and Discharge

By far, the largest number of patients were walk-in patients (315; 79.3%), followed by ambulance transfers (39; 9.8%). The majority of patients were discharged from the ED (334; 84.1%). Fifty-nine patients (14.9%) were hospitalized as in-patients and three (0.8%) patients were transferred to another hospital. One 57-year-old patient was transferred from another hospital by polytrauma and passed away in the ED under reanimation.

With respect to the Swiss Emergency Triage Scale, the majority of patients (303; 76.3%) had urgent triage, followed by emergent (65; 16.4%) triage. Twenty-one patients (5.3%) had such a severe injury that they were triaged as very emergent.

No significant differences were observed between *route of admission*, *route of discharge*, and *triage* in relation to age group, season, or year of consultation.

## 4. Discussion

This study identified 397 patients who sustained occupational injuries while working for a construction income and were covered by Suva insurance, the Swiss Accident Insurance Fund. Consistent with previous studies, our research shows a predominance of male patients [37,38], and is in accordance with statistics showing that persons suffering from these types of injuries are males in 98% of the cases [37]. This is not surprising, considering that males, due to the nature of the job, are more frequently employed in high-risk sectors (e.g., construction industries and manufacturing) compared to females [39]. The most represented age class was 26–35 years old. This is in contrast with other studies demonstrating that young workers sustained a higher risk of work-related injuries due to the shorter duration of occupation and the associated inexperience, less awareness about occupational hazards, and risk-taking behavior [40,41].

Moreover, our study identified a significantly increased risk of work-related injury among construction workers during the warm season (e.g., summer and autumn), which is in accordance with studies from neighboring countries and suggests an association between increased risk of occupational injuries and higher environmental temperatures [27,42,43,44]. However, this may largely be due to more construction activities taking place in the warmer season. In addition, construction workers are at enhanced risk of heat-related injuries due to the handling of heavy machinery, power tools, and heavy workloads, factors which act synergistically with the direct sunlight exposure in causing heat-related illness [45]. However, despite the association between high environmental temperatures and work-related injuries being complex, it is well established that the extreme temperature exposure contributes significantly to reduced productivity, fatigue, carelessness, impaired judgment, poor coordination, loss of concentration, and disorientation increasing the risk of accidental events [46,47,48].

The majority of work-related injuries in our study resulted from a fall (21.9%) and were followed by impact against an object (17.4%) and trauma due to working with manual instruments (17.1%). Our findings are in line with previous evidence which highlights that falls are the leading mechanism of injury among construction workers [49,50,51,52]. Moreover, previous research suggests that human behavior, workplace conditions, safety performance, and the nature of activity might be causal factors for accidents associated with occupational falls [53]. In addition, various parameters such as carelessness, overconfidence, and incorrect evaluation of the height are described as leading causes of work-associated accidents due to falls [54]. Furthermore, other studies have shown that gender and age have an epidemiological association with occupational injuries due to falls [55,56]. In our study, only 16% of accidents occurred in the age group between 16 and 25 years of age, and thus inexperience, risk-taking behavior, and physical vulnerability do not appear as the main contributing factors.

In the present study, the most common body parts injured were the upper extremities, with the hand being the most frequent location of the injury, in accordance with numerous studies describing the upper extremities as the most commonly affected body part [57,58]. Unlike previous evidence indicating lower extremities as the second-most-frequently injured body part [51,59], the results of the present study indicate that it is rather the head, followed by the lower extremities. This finding is concerning, given that a pooled proportions meta-analysis demonstrated that 17.9% of traumatic brain injuries are work-associated and 6.3% of occupational events result in traumatic brain injury (TBI) with 3.6% of work-related TBI being fatal [60]. Interestingly, 77 patients (19.4%) in our study were treated in the ophthalmology area due to eye injury, which is in accordance with numerous studies describing eye injury as a frequent work-related injury representing 30% to 70% of the consultations of the Ophthalmological Emergency Department [61,62,63]. In addition, it is well documented that the majority of patients with work-related eye injuries are construction workers [62,64], indicating the significance of optimizing prevention strategies and safety performance in the construction sector. In our analysis, the most common type of injury was wound lacerations (47.9%), followed by joint distortions (31%) and bone fractures (18.3%). These results confirm previous studies reporting that wound lacerations, sprains and strains, and fractures are frequently observed types of injury among construction workers [23,58,65]. Moreover, as shown in Figure 2, accidents related to the neck/spine, thorax/back, abdomen, and hip resulted in more hospitalization days (approx. up to 10 days) compared to all other body parts. Within these injuries, extreme outliers were present for the neck/spine, thorax/back, and abdomen, presumably indicating the severity of injuries related to the specific body parts. Taking into account that falls are the leading mechanism of injury in the context under examination, protective equipment targeting the extremities and head is strongly suggested.

In contrast with other studies demonstrating a higher risk of occupational injuries during the afternoon and night-shifts [66], two peaks of a daily time of consultations between 06:00 and 12:00 and between 18:00 and 00:00 were observed in our study, indicating that the majority of accidents occurred in the morning- and early-afternoon shifts. A possible explanation for this finding is that construction employees commonly suffer from pain, fatigue, uncontrolled stress, and other special diseases leading to sleep irregularities [67,68,69,70], for example irregular sleep patterns and lower sleep efficiency [71,72]. Moreover, the finding that the number of consultations due to construction accidents between 06:00 and 12:00 is much higher in the warm year period indicates a climate influence. Indeed, extreme environmental temperatures can result in cognitive disturbances such as loss of concentration and disorientation, contributing to the enhanced risk of injury observed in the present study [47]. Additionally, exposure to heat can lead to dehydration and sweaty palms, both serving as risk indicators for occupational injuries [46].

Considering the severity of the injuries, our data show that the proportion of severe injuries is relatively low and comparable to that previously reported [38]. Indeed, only 9.8% of the patients had to be transferred by ambulance, whereas 79.3% were walk-in patients. Moreover, the majority of the patients (76.3%) was assessed as “urgent” triage. Only 5.3% of the patients had such a severe injury requiring immediate examination. In addition, most of the patients received conservative treatment (244 cases; 61.5%), whereas 85 (21.4%) and 56 (14.1%) patients received minimal invasive and operative treatment. Finally, 84.1% of the patients were discharged home, while only 14.9% of the cases had to be hospitalized, and 0.8% were transferred to another hospital.

Despite the strengths of our research, some limitations should be taken into consideration. Specifically, the present study was a single-center study. Injuries related to construction accidents treated in other emergency departments, private physicians’ practices or not treated injuries were not recorded. Moreover, a number of important variables such as level of education and work experience, socioeconomic characteristics, and type of occupation (e.g., permanent, part-time, casual, work in a shift-model) were unfortunately not available. Another limitation was the absence of information of using personal protective equipment and whether safety training had been completed. Furthermore, construction workers who are employed by foreign companies are not covered by Suva and therefore excluded, as well as patients who have not been employed at all, i.e., had an accident on their own, private construction site. Additionally, long-term follow-up data offering supplemental information regarding health costs and persistence of disability were not included in our study. Finally, for some cases data were incomplete or missing.

## 5. Conclusions

Work-related accidents in the construction sector create a globally significant burden of disability and have devastating socioeconomic effects. However, many of them could be preventable by the optimized development of prevention strategies and safety measures. Our results demonstrate that the upper and lower extremities and head are common locations of injuries among construction workers indicating that accurate protective equipment targeting the extremities, eyes, and head is crucial. Moreover, similar to other studies, we identified falls as the leading mechanism of injury leading to strongly recommend primary and secondary improvement of fall prevention. In addition, this study demonstrates that accidental events in the construction sector occur frequently in the warm seasons of the year. Taking into account that the construction sector is highly affected by environmental parameters and that severe weather events are expected to increase due to climate change, it is very important to establish active prevention policies. In order to be conclusive about the possible causes of construction-related accidents more studies are needed focusing on both, local and national levels, and adopting standardized assessment tools. Finally, future studies should additionally strive to track and record cases treated in physicians’ practices and untreated cases to gain more insight into the actual prevalence of these injuries and their possible causes.

## Figures and Tables

**Figure 1 ijerph-19-11294-f001:**
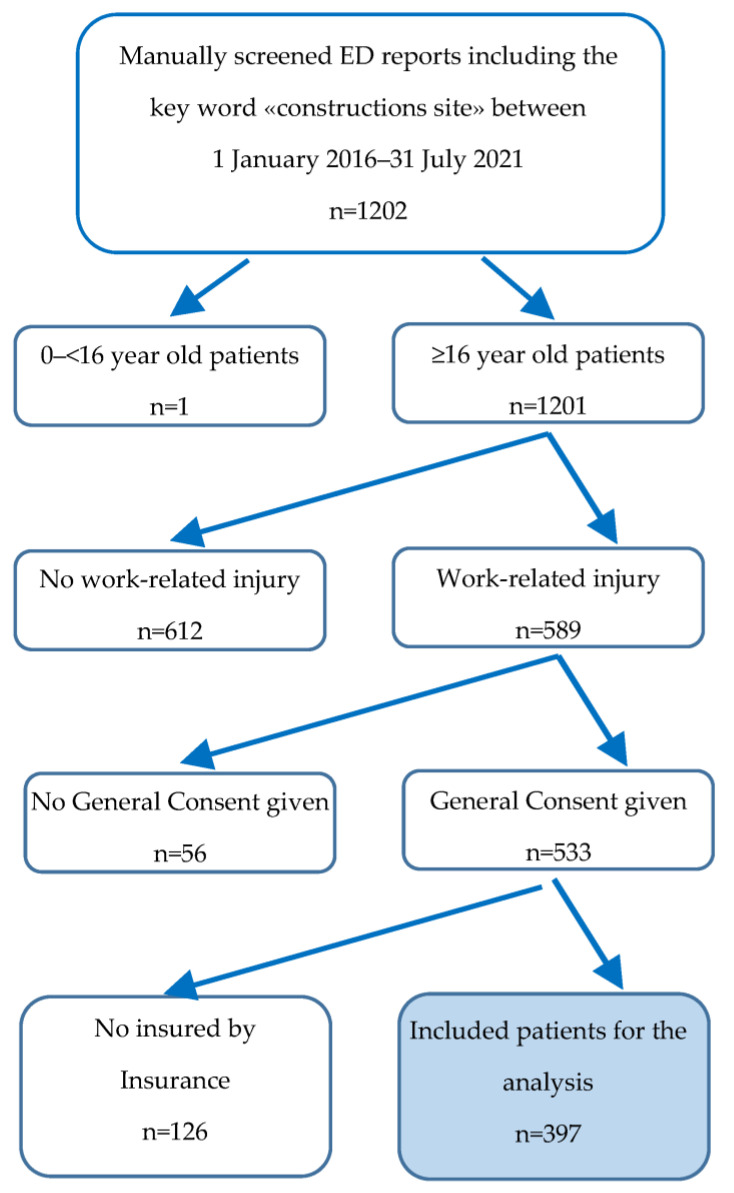
Flow chart of medical record selection.

**Figure 2 ijerph-19-11294-f002:**
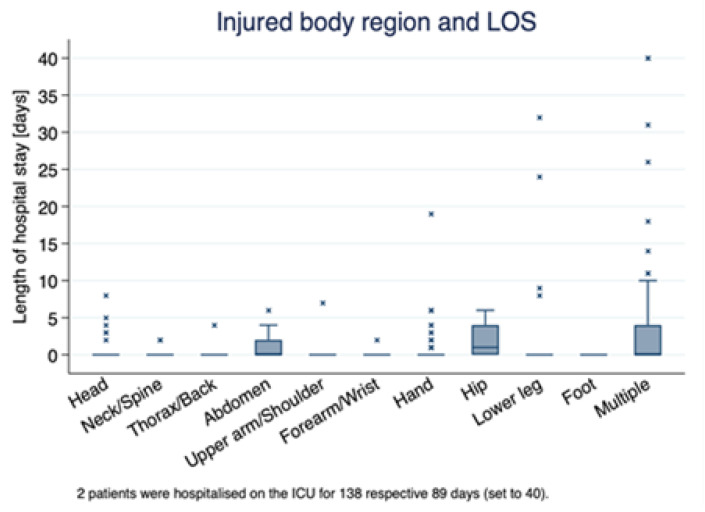
Relationship between the injured body part reported and the length of hospital stay. Outliers are indicated with symbol “×”.

**Figure 3 ijerph-19-11294-f003:**
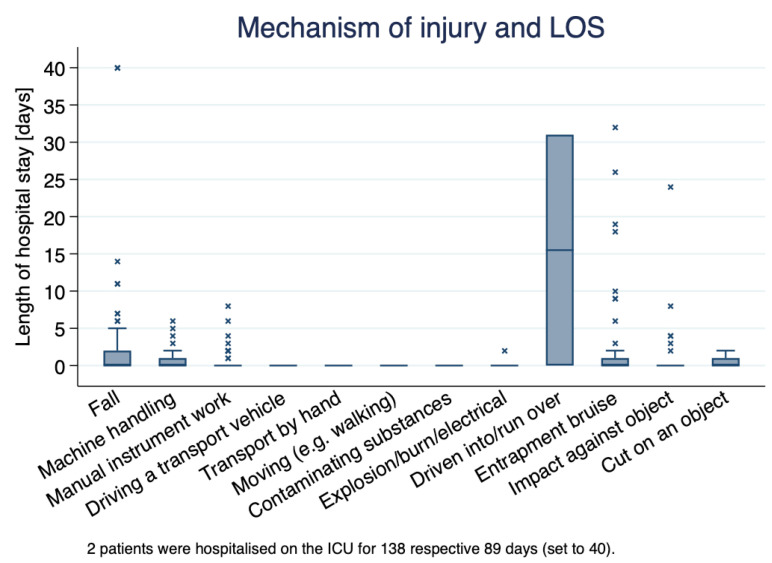
Relationship between the mechanism of injury and length of hospital stay. Outliers are indicated with symbol “×”.

**Figure 4 ijerph-19-11294-f004:**
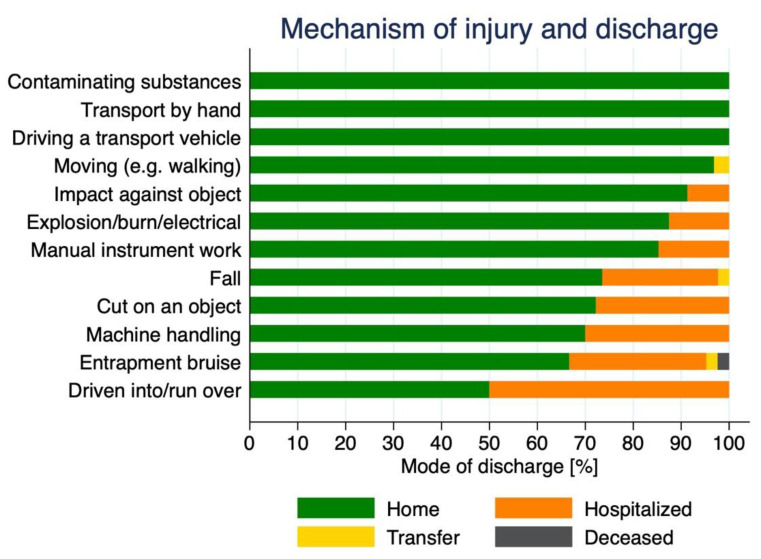
Relation between mechanism of injury and mode of discharge.

**Table 1 ijerph-19-11294-t001:** Distribution of injured body part among construction workers in Switzerland according to age group.

	Total (N = 397)		Age Group										
**Injured body part**	n	(%)	16–25 (n = 80)	(%)	26–35 (n = 107)	(%)	36–45 (n = 98)	(%)	46–55 (n = 74)	(%)	56–65 (n = 38)	(%)	*p*-value
**Head**	136	(34.3)	33	(41.2)	33	(30.8)	33	(33.7)	20	(27.0)	17	(44.7)	0.209
**Head** ** *(single injury)* **	108	(27.2)	27	(33.8)	28	(26.2)	26	(26.5)	16	(21.6)	11	(28.9)	0.552
**Neck, spine**	36	(9.1)	6	(7.5)	8	(7.5)	10	(10.2)	7	(9.5)	5	(13.2)	0.826
**Neck, spine** ** *(single injury)* **	13	(3.3)	2	(2.5)	4	(3.7)	2	(2.0)	3	(4.1)	2	(5.3)	13
**Thorax, back**	30	(7.6)	5	(6.2)	7	(6.5)	10	(10.2)	4	(5.4)	4	(10.5)	0.680
**Thorax, back** ** *(single injury)* **	11	(2.8)	1	(1.2)	3	(2.8)	3	(3.1)	3	(4.1)	1	(2.6)	0.883
**Abdomen, pelvis**	23	(5.8)	6	(7.5)	6	(5.6)	5	(5.1)	3	(4.1)	3	(7.9)	0.872
**Abdomen, pelvis** ** *(single injury)* **	8	(2.0)	1	(1.2)	3	(2.8)	2	(2.0)	1	(1.4)	1	(2.6)	0.937
**Shoulder, upper arm, elbow**	43	(10.8)	10	(12.5)	5	(4.7)	13	(13.3)	9	(12.2)	6	(15.8)	0.189
**Shoulder, upper arm, elbow** ** *(single injury)* **	23	(5.8)	7	(8.8)	1	(0.9)	5	(5.1)	6	(8.1)	4	(10.5)	0.082
**Forearm, wrist, carpus**	27	(6.8)	3	(3.8)	8	(7.5)	6	(6.1)	9	(12.2)	1	(2.6)	0.221
**Forearm, wrist, carpus** ** *(single injury)* **	19	(4.8)	1	(1.2)	8	(7.5)	5	(5.1)	5	(6.8)	0	(0.0)	0.167
**Hand**	92	(23.2)	20	(25.0)	23	(21.5)	20	(20.4)	21	(28.4)	8	(21.1)	0.743
**Hand** ** *(single injury)* **	79	(19.9)	16	(20.0)	19	(17.8)	19	(19.4)	18	(24.3)	7	(18.4)	0.864
**Hip joint, femur**	11	(2.8)	0	(0.0)	3	(2.8)	6	(6.1)	2	(2.7)	0	(0.0)	0.114
**Hip joint, femur** ** *(single injury)* **	8	(2.0)	0	(0.0)	2	(1.9)	5	(5.1)	1	(1.4)	0	(0.0)	0.119
**Knee joint, lower leg**	60	(15.1)	11	(13.8)	23	(21.5)	11	(11.2)	12	(16.2)	3	(7.9)	0.179
**Knee joint, lower leg** ** *(single injury)* **	45	(11.3)	8	(10.0)	17	(15.9)	8	(8.2)	10	(13.5)	2	(5.3)	0.280
**Foot**	21	(5.3)	4	(5.0)	7	(6.5)	7	(7.1)	2	(2.7)	1	(2.6)	0.637
**Foot** ** *(single injury)* **	15	(3.8)	3	(3.8)	4	(3.7)	6	(6.1)	1	(1.4)	1	(2.6)	0.589
**Multiple injured body parts**	58	(14.6)	13	(16.2)	13	(12.1)	14	(14.3)	10	(13.5)	8	(21.1)	0.729

**Table 2 ijerph-19-11294-t002:** Distribution of type of injury among construction workers in Switzerland according to age group.

	Total (N = 397)		Age Group										
**Type of injury**	n	(%)	16–25 (n = 80)	(%)	26–35 (n = 107)	(%)	36–45 (n = 98)	(%)	46–55 (n = 74)	(%)	56–65 (n = 38)	(%)	*p*-value
**Wound laceration, incl. pneumothorax, internal organs**	190	(47.9)	40	(50.0)	54	(50.5)	46	(46.9)	31	(41.9)	19	(50.0)	0.809
**Traumatic brain injury**	32	(8.1)	6	(7.5)	6	(5.6)	14	(14.3)	2	(2.7)	4	(10.5)	0.056
**Distortion**	123	(31.0)	27	(33.8)	40	(37.4)	25	(25.5)	19	(25.7)	12	(31.6)	0.321
**Contusion**	84	(21.2)	19	(23.8)	17	(15.9)	25	(25.5)	18	(24.3)	5	(13.2)	0.275
**Crush trauma**	21	(5.3)	5	(6.2)	4	(3.7)	5	(5.1)	2	(2.7)	5	(13.2)	0.174
**Closed fracture**	49	(12.3)	6	(7.5)	13	(12.1)	12	(12.2)	12	(16.2)	6	(15.8)	0.528
**Open fracture**	24	(6.0)	4	(5.0)	5	(4.7)	5	(5.1)	7	(9.5)	3	(7.9)	0.661
**Dislocation**	6	(1.5)	1	(1.2)	2	(1.9)	2	(2.0)	1	(1.4)	0	(0.0)	0.923
**Burn frostbite**	5	(1.3)	2	(2.5)	1	(0.9)	1	(1.0)	0	(0.0)	1	(2.6)	0.619
**Chemical burn, electric shock, chemical substances**	7	(1.8)	2	(2.5)	3	(2.8)	1	(1.0)	1	(1.4)	0	(0.0)	0.738
**Infection poisoning, thermal shock**	2	(0.5)	0	(0.0)	1	(0.9)	1	(1.0)	0	(0.0)	0	(0.0)	0.756
**Irritation mucous membranes**	10	(2.5)	2	(2.5)	4	(3.7)	4	(4.1)	0	(0.0)	0	(0.0)	0.341
**Penetration and irritation of foreign objects**	71	(17.9)	18	(22.5)	22	(20.6)	13	(13.3)	10	(13.5)	8	(21.1)	0.363
**Amputation**	3	(0.8)	0	(0.0)	0	(0.0)	3	(3.1)	0	(0.0)	0	(0.0)	0.056

**Table 3 ijerph-19-11294-t003:** Distribution of work-related accidents by mechanism of injury among construction workers in Switzerland between 2016 and 2020.

Mechanism of Injury	Number of Patients (Total 397)	
	n	(%)
**Fall**	87	(21.9)
**Machine handling**	40	(10.1)
**Manual instruments**	68	(17.1)
**Driving a transport vehicle**	3	(0.8)
**Transport by hand**	11	(2.8)
**Moving, walking, running, climbing, tripping**	32	(8.1)
**Contaminating substances**	13	(3.3)
**Explosion, ignite burn, electrical contact**	8	(2.0)
**Driven into, run over**	2	(0.5)
**Entrapment bruise**	42	(10.6)
**Impact against object**	69	(17.4)
**Cut on an object**	18	(4.5)

## Data Availability

Not applicable.

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
