# Peer review of "Work-Related Injuries among Insured Construction Workers Presenting to a Swiss Adult Emergency Department: A Retrospective Study (2016–2020)"

_ijerph, 2022, doi:10.3390/ijerph191811294_

Round 1

Reviewer 1 Report

1. The objective of investigate the ep-idemiology of fatal and non-fatal injuries among construction workers older than 16 years of age over a 5-year period. The data source is 397 patients in the emergency department of the University Hospital of Bern. What is the authority of these data? What is the proportion of the 397 patients in the country who are injured and seek medical treatment for construction workers in the country? Can these data be used as reference data for the purpose of writing the article?

2. Figure 1 The picture is blurry and the text is unclear.

3. Accident locations mentioned in line 199 of the article: Different construction sites pose different dangers to workers. Only building construction, road construction and digging construction are mentioned, and electric shock and mechanical injury accidents are suggested to be added.

4. The color contrast in Figure 4 is low, and the visual effect is not very good.

Author Response

Bern, 29. August 2022

Journal IJERPH

Manuscript ID ijerph-1872832

Dear Editor

We appreciate your interest and helpful attentative peer-review of this paper:

Work-related Injuries Among Insured Construction-workers Presenting to a Swiss Adult Emergency

We were pleased to hear that IJERPH is interested in a revised version of our manuscript. We are grateful to the reviewers for their helpful suggestions and comments. All comments have been addressed in the revised version. In more detail:

REVIEWER 1

  1. The objective of investigate the ep-idemiology of fatal and non-fatal injuries among construction workers older than 16 years of age over a 5-year period. The data source is 397 patients in the emergency department of the University Hospital of Bern. What is the authority of these data? What is the proportion of the 397 patients in the country who are injured and seek medical treatment for construction workers in the country? Can these data be used as reference data for the purpose of writing the article?

Reply: We thank the reviewer for the constructive comments and for the opportunity to elaborate on the specific point. A comparison with the SSUV (Sammelstelle für die Statistik der Unfallversicherung, Collection Point for Accident Insurance Statistics) data οf all accidents is indeed possible. The numbers for the main construction trade or for the combined main and construction sub-trade such as production area, workshop, factory, public buildings etc. might be evaluated. About 2/3 of the accidents occurring in these industries actually happen on construction sites. Comparable are for example the age distribution and the distribution on the body parts. In terms of injury types, there may be a larger proportion of fractures at University Hospital of Berne than in statistics of SSUV on accidents in this industry, so accident victims with fractures may have gone to University Hospital of Berne with above-average frequency and, thus, ended up in the study. The other injury types do not significantly deviate from the expected. In any case, it should be borne in mind that the object of the study is to examine the epidemiological data of patients treated for construction injuries at the Emergency Department of the University Hospital of Berne. Patients with minor injuries may not have sought medical attention or they may have been treated at other hospitals or in primary health care, which is included as a limitation in the article.

  1. Figure 1 The picture is blurry and the text is unclear.

Reply: Corrected as suggested

  1. Accident locations mentioned in line 199 of the article: Different construction sites pose different dangers to workers. Only building construction, road construction and digging construction are mentioned, and electric shock and mechanical injury accidents are suggested to be added.

           Reply: We thank the reviewer for bringing this issue to our attention. The specific mechanisms of injury (i.e. electric shock and mechanical injury) are included in the description of the methodology in the category “object causing accidents” as “electricity” and “machines/manual instruments”, respectively. Moreover, in the context of the investigation of the mechanism of the accident the specific mechanisms are mentioned. More specifically, we refer to mechanical injury as “machine handling”, “working with manual instruments”, and to electric shock as “contaminating substances/explosion, burn, electrical”.

  1. The color contrast in Figure 4 is low, and the visual effect is not very good.

Reply: Figure 4 has been changed as recommended

Thank you for the thorough and helpful reviewing and for the opportunity to improve the manuscript. We hope that we have adequately responded to your excellent comments. Having made this revision, we hope that the manuscript would be acceptable for publication in the International Journal of Environmental Research and Public Health in its current version.

We wish you best regards and good health.

-Dr.med. et Dr.phil.  Mairi Ziaka

Critical Care and Internal Medicine physician

Jolanta Klukowska-Rötzler

-Study Director

Reviewer 2 Report

The scope of the study is to investigate the epidemiology of fatal and non-fatal injuries among construction workers older than 16 years over a 5-year period.

The paper approaches a topical issue and the manuscript is clear and presented in a well-structured manner.

The cited references are relevant and most of them are published in the last 5 years.

Specific comments:

1). Row 33 – Introduction section correctly identifies the construction industry as one of the industrial activities with the highest rate of occupational accidents. However, the Introduction section should be completed with information about similar studies performed (if available) and more clarifications about the importance of the study, its scope and significance in the context of the presented facts about the construction industry.

2). Row 34 – The text is aligned to left. Please double-check the alignment of whole manuscript to be justified (text evenly distributed between the margins);

3). Rows 39 – 40 - Please, remove the space between paragraphs and double-check the whole manuscript for this issue (see the Instructions for Authors, available online on the IJERPH website);

4). Rows 73 – 77 – Please, explain how you established the 5 years period and why you consider it as being relevant for identifying trends, causes, risk factors, underlying mechanisms, and trauma patterns;

5). Rows 79, 85, 92, 114, 174, 183, 186 – Please, consider to transform the titles to subsections, numbered with 2.1., 2.2. and so on (see the template information in Instructions for Authors, available online on the IJERPH website);

6). Rows 93, 96 – Please, consider to transform the titles to subsubsections (e.g. numbered 2.3.1. and 2.3.2.) - see the template information in Instructions for Authors, available online on the IJERPH website;

7). Rows 118 – 169 – Please, consider to organise the information in a table, for a better readability;

8). Rows 366 – 374 – Information about Personal Protective Equipment (e.g. its availability, quality, correctly using by workers) and information about Occupational Health and Safety training (its availability and quality) are also important variables not considered in the study and should be mentioned as limitations;

9). Row 405 – Reference 3 seems to be incomplete, please double-check it.

Author Response

Bern, 29. August 2022

Journal IJERPH

Manuscript ID ijerph-1872832

Dear Editor

We appreciate your interest and helpful attentative peer-review of this paper:

Work-related Injuries Among Insured Construction-workers Presenting to a Swiss Adult Emergency

We were pleased to hear that IJERPH is interested in a revised version of our manuscript. We are grateful to the reviewers for their helpful suggestions and comments. All comments have been addressed in the revised version. In more detail:

REVIEWER 2

Comments and Suggestions for Authors

The scope of the study is to investigate the epidemiology of fatal and non-fatal injuries among construction workers older than 16 years over a 5-year period.

The paper approaches a topical issue and the manuscript is clear and presented in a well-structured manner.

The cited references are relevant and most of them are published in the last 5 years.

Reply: We thank the reviewer for her/his comments

Specific comments:

1). Row 33 – Introduction section correctly identifies the construction industry as one of the industrial activities with the highest rate of occupational accidents. However, the Introduction section should be completed with information about similar studies performed (if available) and more clarifications about the importance of the study, its scope and significance in the context of the presented facts about the construction industry.

Reply: We thank the reviewer for bringing this issue to our attention. In light of the edifying recommendation additional information about previous studies is now added in the introduction section (kindly see page 2, 1st  paragraph). Moreover, in order to complete the introduction section, further observations based on previous research about mechanism of injury and clinical characteristics are now stated in the manuscript (kindly see page 2, 3rd paragraph). In addition, the importance of the study, its scope, and significance has been highlighted in the last paragraph of the introduction section.

2). Row 34 – The text is aligned to left. Please double-check the alignment of whole manuscript to be justified (text evenly distributed between the margins);

Reply: Done as recommended

3). Rows 39 – 40 – Please, remove the space between paragraphs and double-check the whole manuscript for this issue (see the Instructions for Authors, available online on the IJERPH website);

Reply: Done as recommended

4). Rows 73 – 77 – Please, explain how you established the 5 years period and why you consider it as being relevant for identifying trends, causes, risk factors, underlying mechanisms, and trauma patterns;

Reply: We chose the 5-year time interval, based on the design of previous similar epidemiological studies, which examined occupational accident data over a period of 3-6 years [1-4].

5). Rows 79, 85, 92, 114, 174, 183, 186 – Please, consider to transform the titles to subsections, numbered with 2.1., 2.2. and so on (see the template information in Instructions for Authors, available online on the IJERPH website);

Reply: Done as recommended

6). Rows 93, 96 – Please, consider to transform the titles to subsubsections (e.g. numbered 2.3.1. and 2.3.2.) - see the template information in Instructions for Authors, available online on the IJERPH website;

Reply: Done as recommended

7). Rows 118 – 169 – Please, consider to organise the information in a table, for a better readability;

Reply: Thank you for bringing the above issue in our attention. We would prefer to retain the text, as we believe that the terms are too long for a table of good quality and may affect readability. However, in case of disagreement we are prepared to organize the information in a table.

8). Rows 366 – 374 – Information about Personal Protective Equipment (e.g. its availability, quality, correctly using by workers) and information about Occupational Health and Safety training (its availability and quality) are also important variables not considered in the study and should be mentioned as limitations;

Reply: We thank the reviewer for the very important comment. In light of the edifying recommendation an additional limitation about lack of information related to personal protective equipment and occupational health- and safety training is now added in the limitations section. 

9). Row 405 – Reference 3 seems to be incomplete, please double-check it.

Reply: We have now completed reference.

References

  1. Sharwood LN, Mueller H, Ivers RQ, Vaikuntam B, Driscoll T, Middleton JW. The Epidemiology, Cost, and Occupational Context of Spinal Injuries Sustained While 'Working for Income' in NSW: A Record-Linkage Study. Int J Environ Res Public Health. 2018, 27;15(10):2121.
  2. Win KN, Trivedi A, Lai A, Hasylin H, Abdul-Mumin K. Non-fatal occupational accidents in Brunei Darussalam. Ind Health. 2021, 17;59(3):193-200.
  3. Unsar, S.; Sut, N. General assessment of the occupational accidents that occurred in Turkey between the years 2000 and 2005. Saf. Sci. 2009, 47, 614–619.
  4. Fabiano, B.; Currò, F.; Reverberi, A.P.; Pastorino, R. A statistical study on temporary work and occupational accidents: Specific risk factors and risk management strategies. Saf. Sci. 2008, 46, 535–544.

Thank you for the thorough and helpful reviewing and for the opportunity to improve the manuscript. We hope that we have adequately responded to your excellent comments. Having made this revision, we hope that the manuscript would be acceptable for publication in the International Journal of Environmental Research and Public Health in its current version.

We wish you best regards and good health.

-Dr.med. et Dr.phil.  Mairi Ziaka

Critical Care and Internal Medicine physician

Jolanta Klukowska-Rötzler

-Study Director

Round 2

Reviewer 1 Report

Agree to publish